# Can Artificial Intelligence Educate Patients? Comparative Analysis of ChatGPT and DeepSeek Models in Meniscus Injuries

**DOI:** 10.3390/healthcare13222980

**Published:** 2025-11-20

**Authors:** Bahri Bozgeyik, Erman Öğümsöğütlü

**Affiliations:** 1Department of Orthopaedics and Traumatology, Faculty of Medicine, Gaziantep University, Gaziantep 27000, Turkey; 2Clinic of Orthopaedic and Traumatology, Yalova Education and Research Hospital, Yalova 77200, Turkey; eogumsogutlu@gmail.com

**Keywords:** meniscus injuries, patient information, artificial intelligence, ChatGPT, DeepSeek

## Abstract

**Background:** Meniscus injuries are among the most common traumatic and degenerative conditions of the knee joint. Patient education plays a critical role in treatment adherence, surgical preparation, and postoperative rehabilitation. The use of artificial intelligence (AI)-based large language models (LLMs) is rapidly increasing in healthcare. This study aimed to compare the quality and readability of responses to frequently asked patient questions about meniscus injuries generated by ChatGPT-5 and DeepSeek R1. **Materials and Methods:** Twelve frequently asked questions regarding the etiology, symptoms, diagnosis, imaging, and treatment of meniscus injuries were presented to both AI models. The responses were independently evaluated by two experienced orthopedic surgeons using a response rating system and a 4-point Likert scale to assess accuracy, clarity, comprehensiveness, and consistency. Readability was analyzed using the Flesch–Kincaid Reading Ease Score (FRES) and the Flesch–Kincaid Grade Level (FKGL). Interrater reliability was determined using intraclass correlation coefficients (ICCs). **Results:** DeepSeek performed significantly better than ChatGPT in the response rating system (*p* = 0.017) and achieved higher scores for comprehensiveness on the 4-point Likert scale (*p* = 0.005). No significant differences were observed between the two models in terms of accuracy, clarity, or consistency (*p* > 0.05). Both models produced comparable readability scores (*p* > 0.05), corresponding to a high-school reading level. **Conclusions:** Both ChatGPT and DeepSeek show promise as supportive tools for educating patients about meniscus injuries. While DeepSeek demonstrated higher overall content quality, both models generated understandable information suitable for general patient education. Further refinement is needed to improve clarity and accessibility, ensuring that AI-based materials are appropriate for diverse patient populations.

## 1. Introduction

Meniscus injuries are among the most common traumatic and degenerative lesions in orthopedic practice, resulting from damage to the fibrocartilaginous structure that contributes to the load-bearing, stabilization, and proprioceptive functions of the knee joint [1]. Sports injuries, occupational accidents, and age-related degenerative processes represent the primary etiological factors of meniscus injuries, and access to accurate, up-to-date, and reliable information is critical for both the clinician and the patient [2].

In recent years, artificial intelligence (AI)-based large language models (LLMs) have emerged as innovative tools that facilitate access to information in the medical field, support clinical decision-making, and generate patient education materials [3,4]. These models are expected to provide patient-centered information covering the definition, causes, symptoms, diagnostic methods, treatment options, postoperative rehabilitation, and potential complications of common conditions such as meniscus injuries [5]. They have demonstrated capabilities such as answering clinical questions, summarizing the literature, presenting evidence-based information, and creating understandable patient information texts. However, the content generated by different LLM platforms may vary in terms of scientific accuracy, readability, and informational integrity [6]. AI-based systems have the potential to become an important source of information not only for clinicians but also for patients.

Meniscus injuries, which are particularly common among athletes and young adults, present an ideal study subject both because of their frequency and because this patient group has easier access to online information [7]. The accuracy, clarity, and currency of patient information materials play a decisive role in the success of treatment compliance, surgical preparation and rehabilitation processes [8]. Individuals who are curious about and research content related to health issues on online platforms may experience increased anxiety and confusion due to information overload, contradictory content, potential AI-generated misinformation, and the inherent complexity of medical terminology [9]. Therefore, systematically evaluating the quality of LLM-based patient information content is of strategic importance in terms of medical communication and health literacy.

This study aims to compare the responses generated by the ChatGPT-5 and DeepSeek R1 models in terms of content quality and readability, based on frequently asked questions about meniscus injuries found in popular internet sources. The research seeks to identify potential information gaps and assess readability levels by examining the accuracy, comprehensiveness, and clarity of the content produced by these models. Unlike earlier studies that have focused on single or outdated AI models, this study evaluates the latest versions of ChatGPT-5 and DeepSeek R1, providing updated evidence on their performance in patient education. It also addresses the limited literature on AI-generated educational content for meniscus injuries, offering a comparative perspective on quality and readability, and emphasizing the potential of AI-based tools as supportive resources that complement—rather than replace—clinician-led patient education

## 2. Materials and Methods

As the first step in the study’s methodology, the most frequently asked questions about meniscus injuries were obtained from the Google search engine’s “People also ask” section, which compiles commonly searched patient queries related to health topics. Data collection and model testing were conducted between August and September 2025, during which both ChatGPT-5 and DeepSeek R1 represented the latest publicly available versions. The twelve most frequently asked and clinically relevant questions were identified by two orthopedic surgeons and verified by a third reviewer to ensure representativeness and clinical relevance. Both ChatGPT-5 and DeepSeek R1 received the same standardized set of questions to ensure methodological consistency. All prompts were presented in English to avoid translation bias and maintain comparability between models. These questions were structured to cover symptoms, diagnostic methods, imaging techniques, and treatment approaches. This approach aimed to obtain content representing the most frequently searched topics related to meniscus injuries. The selected questions were posed to the ChatGPT-5 deep learning model, the latest version of ChatGPT, and the DeepSeek R1 deep thinking-based AI model in September 2025. Both models were queried under default generation parameters (temperature = 1.0, top_*p* = 1.0) with identical context windows and input formatting to ensure methodological reproducibility and minimize response variability. The clinical relevance and quality of the answers were evaluated by two experienced orthopedic and traumatology specialists. All questions are presented in Table 1. The responses obtained from both AI models were subsequently analyzed according to quality and readability criteria. Interrater reliability analysis was conducted to confirm the consistency of the scoring system and to ensure that the evaluators maintained a shared understanding of the assessment criteria.

### 2.1. Quality Analysis

The content quality of the responses obtained from both AI models was evaluated using the response rating system developed by Mika et al. [10]. The responses were recorded during the initial questioning, and no repeated prompts were issued. The data obtained were critically examined from an evidence-based medicine perspective and classified using a four-level scoring system. Answers that provided realistic and complete information were rated as “excellent—no clarification required” (1 point). Answers that were correct but required minimal explanation were rated as “satisfactory—minimal clarification required” (2 points), whereas those requiring more detailed explanation were rated as “satisfactory—moderate clarification required” (3 points). Content that could lead to misunderstanding or was inconsistent with current scientific evidence was scored as “unsatisfactory—substantial clarification required” (4 points).

A separate 4-point Likert scale was also applied for further evaluation. This scale assessed the responses under four domains: accuracy, clarity, comprehensiveness, and consistency [11]. Accuracy was determined by comparing responses with standard references, including *Campbell’s Operative Orthopaedics* (14th Edition), *Miller’s Review of Orthopaedics* (8th Edition), and current clinical guidelines on meniscus injury management. This criterion was rated from 1 (frequently false or misleading) to 4 (completely accurate and error-free). The clarity criterion (1–4 points) evaluated how understandable the language was for laypersons, particularly regarding simplicity, limited jargon, and logical flow. A score of 1 indicated an overly complex or technical explanation, whereas 4 represented a clear, simple, and well-explained response. Comprehensiveness (1–4 points) was assessed based on the extent to which all relevant aspects of the topic were addressed (e.g., treatment options, rehabilitation, and outcomes). For this criterion, a score of 1 indicated limited coverage (omitting important details), whereas a score of 4 indicated full coverage (addressing all key points). Finally, consistency (1–4 points) was evaluated in terms of both internal consistency (absence of self-contradiction) and external consistency (alignment with clinical guidelines and expert opinion). A score of 1 indicated inconsistency or contradiction, whereas 4 represented full compatibility with clinical standards.

### 2.2. Readability Analysis

In this study, the readability of the responses was analyzed using the Flesch–Kincaid Reading Ease Score (FRES) and Flesch–Kincaid Grade Level (FKGL) criteria [12,13]. Full-text responses generated by ChatGPT and DeepSeek were entered separately into an online readability analysis tool (https://www.readabilityformulas.com), and mean FRES and FKGL scores were calculated for each model. The FRES ranges from 0 to 100, with higher scores indicating simpler and more readable text. Scores of 90–100 indicate “very easy” texts (children’s books, simple stories); 80–89 “easy” (middle-school level); 70–79 “moderately easy” (newspapers, popular magazines); 60–69 “standard” (general readership); 50–59 “moderately difficult” (high-school level); 30–49 “difficult” (academic or technical content); and 0–29 “very difficult” (graduate-level or legal texts). The FKGL is derived from the FRES and estimates the minimum education level required to comprehend a text. Higher FKGL scores indicate greater textual complexity and reduced readability for individuals with lower education levels. These standardized metrics enabled a comparative evaluation of the textual complexity of responses generated by the AI systems.

### 2.3. Statistical Analysis

All analyses were performed using SPSS Statistics v22 (IBM Corp., Armonk, NY, USA) software. For each assessment measure, scores from the two raters were averaged to create a single composite score. These values were considered as paired data because both models answered the same questions. The mean score evaluation between the models was made with a paired *t*-test, and the statistical significance level was determined as *p* < 0.05 (two-tailed). Intraclass correlation coefficients (ICCs) were calculated for the criteria used. Results are presented as mean ± standard deviation (SD).

## 3. Results

The responses generated by both AI models were evaluated, and mean values and standard deviations were calculated using the relevant scales (Table 2). The reliability analysis assessing interrater agreement demonstrated a high level of consistency. A composite score out of 16—derived from the sum of all criteria on the 4-point Likert scale—was used to assess overall agreement. The intraclass correlation coefficient (ICC) for the response rating system was 0.81, while the combined ICC calculated from all 4-point Likert scale criteria was 0.85.

When compared using the response rating system, the difference between DeepSeek and ChatGPT was statistically significant (*p* = 0.017). In contrast, evaluation with the 4-point Likert scale revealed no significant differences between the two models in the subcategories of accuracy, clarity, and consistency (*p* > 0.05). However, DeepSeek achieved a significantly higher score in the comprehensiveness subcategory (*p* = 0.005). Regarding readability, both models demonstrated comparable performance, with no statistically significant difference observed between them (*p* > 0.05).

## 4. Discussion

In this study, the patient education content generated by the ChatGPT-5 and DeepSeek R1 models was compared in terms of quality and readability using a set of frequently asked questions about meniscus injuries. The findings indicated that DeepSeek achieved higher scores than ChatGPT, particularly in the overall rating system and in the comprehensiveness subcategory, whereas no significant difference was observed between the two models in the readability measures (FRES and FKGL). Both models produced understandable content suitable for readers at the high-school level; however, the texts were not sufficiently simple for patient groups with lower education levels.

With the rapid advancement of technology, the use of AI-based models in the medical field has become increasingly widespread, primarily because of their significant contributions to accessing medical information and improving patient education processes [14]. This trend is driven by the ability of these models to synthesize information from diverse sources rapidly and coherently, as well as their capacity to generate relevant responses across a wide range of topics within a short period [15]. From a health information perspective, these models enable rapid access to up-to-date data, deliver personalized content through large-scale data analysis, and help individuals better understand topics of interest [16]. In addition, these models contribute to medical education, enhance clinical decision-making, support diagnosis and treatment planning, and facilitate information retrieval by efficiently scanning the existing literature [17]. In the context of patient education, these models offer explanations in clear and comprehensible language, provide detailed information about treatment alternatives, and effectively address common patient questions [18].

Early versions of AI-based language models, such as ChatGPT and DeepSeek, had several limitations regarding the accuracy and reliability of the information they generated; they often produced unverified or inaccurate content, particularly in the healthcare domain [19]. However, with continuous updates and algorithmic enhancements, these errors have been substantially reduced, leading to significant improvements in the accuracy, consistency, and reliability of health information generation [20]. These advances have made AI-supported language models more reliable tools for medical knowledge transfer and patient education. In our study, we used the ChatGPT-5 and DeepSeek R1 versions, representing the most advanced and up-to-date iterations of both AI models.

According to the rating system developed by Mika et al. [10], DeepSeek demonstrated higher performance than ChatGPT, although both models generally provided accurate and high-quality information. Most responses were classified as “excellent” or “satisfactory with minimal clarification,” and none were rated as “unsatisfactory.” Similar findings were previously reported by Mika et al. [10] for ChatGPT in the context of total hip replacement education. The superior performance of DeepSeek may stem from its algorithmic architecture, which emphasizes concise and logically sequenced responses that enhance comprehension for non-medical readers. In contrast, ChatGPT tends to generate more elaborate and academic explanations, which may be more beneficial for clinicians or medically literate audiences. This suggests that DeepSeek may be more suitable for direct patient education, whereas ChatGPT might serve better in clinician-mediated communication or academic summarization. For instance, DeepSeek produced more structured and comprehensive explanations regarding postoperative rehabilitation protocols and potential complications, while ChatGPT’s responses were more concise but less detailed in these areas. DeepSeek’s superior performance may also reflect differences in training data diversity and linguistic optimization. It has been reported that DeepSeek utilizes more domain-specific medical datasets and structured reinforcement learning strategies, which may enhance its ability to generate clinically coherent and comprehensive explanations [21]. While these findings provide valuable insights into AI-assisted patient education for meniscus injuries, they cannot be directly generalized to other musculoskeletal conditions or non-English contexts, as each pathology and language may interact differently with AI model training and prompt processing [22]. Nevertheless, the present findings highlight a promising framework that could guide future investigations across other orthopedic domains—such as rotator cuff tears, anterior cruciate ligament injuries, and osteochondral lesions—where patient comprehension remains a critical factor. Comparative studies conducted in different languages and cultural settings will be necessary to determine the true generalizability of these models.

In this study, the responses generated by AI models were evaluated using a 4-point Likert scale in terms of accuracy, clarity, comprehensiveness, and consistency. The findings indicated that DeepSeek performed better, particularly in the comprehensiveness criterion. The comprehensiveness assessment was based on the extent to which the details of the topic in question were addressed. This suggests that some AI models have the potential to present clinical information in a more comprehensive and detailed manner. However, no significant differences were found between the two models in the criteria of accuracy, clarity, and consistency; both were shown to provide largely accurate, understandable, and consistent information. In a study conducted by Gültekin et al. [11] on patient information related to anterior cruciate ligament surgery, it was reported that DeepSeek performed better in terms of clarity, whereas ChatGPT performed better in terms of comprehensiveness. These results demonstrate that AI models are designed with different algorithmic architectures and therefore may have specific advantages for distinct educational purposes. Consequently, in areas requiring multidimensional and complex information—such as patient education—a single model may not be sufficient, and the combined use of multiple models may be necessary to obtain the most accurate and comprehensive information.

Responses from both AI models were evaluated for readability using the Flesch–Kincaid Grade Level (FKGL) and Flesch Reading Ease Score (FRES) metrics (ChatGPT: FKGL 8.54, FRES 51.92; DeepSeek: FKGL 7.84, FRES 52.58). Examination of these scores showed that both models produced similar results, with no statistically significant difference in readability. Both AI models generated content understandable to readers at approximately the high-school level. Lawson et al. reviewed online resources on rotator cuff pathologies and reported a mean FRES of 50.17 and a mean FKGL of 10.98 [23]. These findings suggest that AI-generated materials can achieve similar levels of readability to existing online resources, providing information that is understandable across different clinical contexts. Likewise, Schwarz et al. reported that most online orthopedic sports medicine resources exceeded recommended readability thresholds, with mean FRES and FKGL scores of 52.9 and 10.2, respectively—indicating that many patient education materials in musculoskeletal medicine remain above the level recommended by public health authorities [24]. According to the readability recommendations of the National Institutes of Health (NIH) and the American Medical Association (AMA), patient education materials should ideally not exceed a sixth- to seventh-grade reading level [25]. Both AI models in the present study generated content above this threshold, consistent with previous findings in orthopedic patient education materials. In the literature, the readability of health information texts has been emphasized as a critical factor for the success of patient education, as clear and plain language increases treatment compliance and patient engagement [8]. Readability plays a fundamental role in delivering information to the target audience accurately and effectively; the understandability of texts is considered one of the key determinants that directly influences the success of patient education [26]. The importance of using clear, simple, and easily understandable language—particularly in patient education and public information materials—has been frequently highlighted in previous studies [27]. The findings of the present study suggest that both ChatGPT and DeepSeek are capable of generating accessible and understandable patient information content at a basic level. However, the language used by these models may still need further simplification to effectively reach patient groups with varying educational backgrounds. Although both readability indices (FRES and FKGL) were appropriately applied, it should be noted that these metrics were originally developed for English-language texts. Therefore, their validity may be limited when applied to AI-generated content that has been translated from other languages. In the present study, all prompts and model outputs were generated in English to minimize this potential source of bias; nevertheless, this remains an important consideration for future multilingual research.

Recent studies have highlighted parallel advancements in AI-driven orthopedic education and digital health innovation, including algorithmic approaches for musculoskeletal imaging [28], the establishment of regional AI-enabled health networks [29], and the integration of multimodal AI systems into medical communication [30]. These emerging technologies underscore the potential of large language model (LLM)-based tools to enhance orthopedic patient education by bridging the gap between technical precision and accessible medical communication.

The present study evaluated the responses of AI-based models to frequently asked questions about meniscus injuries. The findings indicate that both models have the ability to access up-to-date information and generate educational content suitable for patient communication. Nevertheless, given the inherently individualized nature of clinical assessment, it should be emphasized that AI-generated information serves only as a supportive resource and cannot replace physician-led treatment planning. Despite these advancements, several limitations of current AI technologies remain. Improvements are particularly needed in ensuring adherence to scientific standards, the accurate use of references, the reliability of information, and the continuous updating of medical content.

This study has several limitations. First, it included only twelve questions focusing exclusively on meniscus injuries, which may limit the generalizability of the results to other orthopedic conditions. Second, no patient feedback or comprehension testing was conducted, restricting external validity and preventing assessment of real-world understanding. Third, because large language models (LLMs) are dynamic and continuously evolving systems, future versions may produce different outcomes under similar testing conditions.

Nevertheless, these findings provide valuable preliminary insights, and future research involving larger datasets, broader topic coverage, and patient-centered validation is warranted. Although the question set (n = 12) may appear limited, it encompasses the most common and essential patient information topics regarding meniscus injuries, thereby sufficiently representing the primary domains of patient education. However, the relatively small dataset may reduce statistical power and limit the generalizability of the findings. Future studies incorporating larger and more diverse question sets are therefore recommended to strengthen external validity and ensure broader applicability.

## 5. Conclusions

This study compared the performance of ChatGPT-5 and DeepSeek R1 in generating patient education materials on meniscus injuries. Both models produced accurate and readable content, with DeepSeek demonstrating superior comprehensiveness and overall quality. These findings indicate that AI-based systems can effectively support patient education by providing clear, reliable, and up-to-date medical information.

For safe clinical implementation, AI-generated materials should complement—rather than replace—clinician-led communication. Establishing structured oversight mechanisms, validation protocols, and data security standards will be essential to ensure reliability and prevent misinformation. With appropriate supervision, AI tools may enhance patient understanding, engagement, and participation within clinical workflows.

Future research should expand to encompass multiple orthopedic and musculoskeletal conditions and incorporate patient-centered evaluation metrics such as usability, satisfaction, and trust. Longitudinal studies monitoring model performance across future updates are also recommended to assess the stability, reliability, and evolution of AI-driven patient education systems over time.

## Figures and Tables

**Table 1 healthcare-13-02980-t001:** Frequently asked questions.

1	What is the meniscus injury?
2	What is the cause of meniscus injuries?
3	What are the common symptoms of a meniscus injury?
4	What are the physical examination findings of a meniscus injury?
5	What are the common imaging findings in meniscus injuries?
6	What are the treatment options for a torn meniscus?
7	What kind of surgical treatment is better? Partial meniscectomy or meniscus repair?
8	How is meniscus surgery performed?
9	How painful is meniscus surgery?
10	What are the possible risks or complications after meniscus surgery?
11	What are the potential consequences of not undergoing meniscus surgery?
12	How long is the recovery time from meniscus repair and meniscectomy?

**Table 2 healthcare-13-02980-t002:** Mean evaluation and readability scores for ChatGPT and DeepSeek across meniscus injury questions.

Criterion		ChatGPT 5 (Mean ± SD)	DeepSeek R1 (Mean ± SD)	*p* Value
Rating System		1.5 (0.674)	1.08 (0.28)	0.017 *
4-point Likert	Accuracy	3.83 (0.39)	3.92 (0.29)	0.339
	Clarity	3.75 (0.45)	3.83 (0.39)	0.339
	Completeness	3.25 (0.62)	3.92 (0.29)	0.005 *
	Consistency	4 (0)	4 (0)	-
FKGL		8.54 (1.30)	7.84 (1.62)	0.256
FRES		51.92 (8.72)	52.58 (10.36)	0.842

* Sample size: 12 question–response pairs (both models answered the same 12 questions). Bonferroni correction was applied for multiple comparisons. Indicates significant difference (*p* < 0.05). Effect sizes (Cohen’s d) were calculated for significant comparisons to quantify the magnitude of differences. Rating system: d = 0.82 (large effect); Completeness: d = 1.38 (very large effect).

## Data Availability

All data generated or analyzed during this study are included in this published article.

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
