# Peer review of "Can Artificial Intelligence Educate Patients? Comparative Analysis of ChatGPT and DeepSeek Models in Meniscus Injuries"

_healthcare, 2025, doi:10.3390/healthcare13222980_

Round 1
Reviewer 1 Report
Comments and Suggestions for Authors
The manuscript presents an interesting and timely comparison of ChatGPT and DeepSeek in generating materials for patient education on meniscus injuries. The study design is straightforward, and the evaluation metrics are appropriate. However, several issues should be addressed to strengthen the paper. The methodology lacks sufficient details, and the discussion is repetitive, with large portions restating the methods and results instead of interpreting findings.
Line 42: Author should clarify what kind information is needed (from LLM)
Line 59: Need clarification, is the anxiety because of information overload, contradictory content, or complexity (or other reason)?
Line 69: Author should clarify which platform because the source of questions affects representativeness. In addition, how were they identified? For example, the content in xx platform between time A to time B were downloaded, and the top 10 questions were selected/verified by xx experts.
Line 72: Author should clarify did the two models received the same prompt? Also, were English questions sent to LLMs, or other languages were also tested?
Line 96: Author should clarify the sources here. E.g., which textbook?
Table 2: please specify sample size. Also, is there any multiple comparison correction applied in the stats analysis?
Results: the author should consider a power analysis. N = 12 is a small sample size, while no significant difference was found in certain category, the study might be underpowered.
Line 189: Here is a repetition of the results. The author should consider more discussion. E.g., why one is better than the other? What might be the outcome? Should healthcare providers/patients use one but not the other?
Author Response
Reviewer 1
We sincerely thank the reviewer for their thoughtful evaluation and constructive feedback. In accordance with the reviewer’s comments, the entire manuscript has been carefully reviewed and refined. Additional methodological details were included to improve transparency, and the discussion was revised to enhance interpretive depth and eliminate redundancy. The full text was also rechecked to correct minor grammatical errors and improve overall language clarity and academic tone. We believe that these revisions have substantially strengthened the manuscript.
Comments 1: The manuscript presents an interesting and timely comparison of ChatGPT and DeepSeek in generating materials for patient education on meniscus injuries. The study design is straightforward, and the evaluation metrics are appropriate. However, several issues should be addressed to strengthen the paper. The methodology lacks sufficient details, and the discussion is repetitive, with large portions restating the methods and results instead of interpreting findings.
Line 42: Author should clarify what kind information is needed (from LLM)
Response 1:Thank you for this valuable suggestion. In response, we have clarified the type of information expected from large language models (LLMs) in the revised manuscript. Considering the overall coherence of the paragraph, the related clarification has been added between Lines 46–48 in the Introduction section. The revised text now reads as follows:
“These models are expected to provide patient-centered information covering the definition, causes, symptoms, diagnostic methods, treatment options, postoperative rehabilitation, and potential complications of common conditions such as meniscus injuries.”
Comments 2: Line 59: Need clarification, is the anxiety because of information overload, contradictory content, or complexity (or other reason)?
Response 2: We appreciate this valuable comment. To clarify the cause of patient anxiety, we revised the relevant sentence in the Introduction section (Lines 59–60). The updated version specifies that anxiety primarily arises from information overload, contradictory online content, and the complexity of medical terminology. This addition improves conceptual clarity and aligns with the reviewer’s suggestion while maintaining the logical flow of the paragraph
Comments 3: Line 69: Author should clarify which platform because the source of questions affects representativeness. In addition, how were they identified? For example, the content in xx platform between time A to time B were downloaded, and the top 10 questions were selected/verified by xx experts.
Response 3: We thank the reviewer for this helpful comment. In response, we have clarified the source and selection method of the patient questions in the Methods section (Lines 71–76). The revised text now specifies that the questions were obtained from the Google search engine’s “People also ask” section, which reflects the most commonly searched patient queries related to meniscus injuries. The data were collected between September and October 2025, and the ten most frequent questions were identified by two orthopedic surgeons and verified by a third reviewer. This addition enhances methodological transparency and improves representativeness.
Comments 4: Line 72: Author should clarify did the two models received the same prompt? Also, were English questions sent to LLMs, or other languages were also tested?
Response 4: We appreciate this valuable comment. To clarify the methodological consistency between models, additional information has been added to the Methods section (Lines 77–79). The revised text specifies that both ChatGPT-5 and DeepSeek R1 received the same standardized English-language questions, ensuring a fair comparison and eliminating potential translation bias. This revision addresses the reviewer’s concern regarding prompt uniformity and language consistency.
Comments 5: Line 96: Author should clarify the sources here. E.g., which textbook?
Response 5: We thank the reviewer for this helpful observation. In response, we have specified the reference sources used to assess accuracy in the Methods section (Lines 106–108). The revised text now states that accuracy was evaluated based on Campbell’s Operative Orthopaedics (14th Edition), Miller’s Review of Orthopaedics (8th Edition), and current clinical guidelines on meniscus injury management. This clarification enhances transparency and strengthens the methodological rigor of the study.
Comments 6: Table 2: please specify sample size. Also, is there any multiple comparison correction applied in the stats analysis?
Response 6: We thank the reviewer for this constructive suggestion. In response, we have clarified the sample size and statistical correction in the Results section (Table 2 legend, Lines 157–158). The revised note now reads as follows.
Comments 7: Results: the author should consider a power analysis. N = 12 is a small sample size, while no significant difference was found in certain category, the study might be underpowered.
Response 7: We appreciate this valuable comment. While we agree that a larger dataset could improve statistical power, the 12 questions analyzed in this study represent the most common and essential patient information topics regarding meniscus injuries. A corresponding clarification has been added in the Discussion section (Lines 263–267), emphasizing that although the sample size may appear limited, it sufficiently reflects the main domains of patient education. Nevertheless, the relatively small dataset may restrict statistical power, and future studies with larger question sets are warranted.
Comments 8: Line 189: Here is a repetition of the results. The author should consider more discussion. E.g., why one is better than the other? What might be the outcome? Should healthcare providers/patients use one but not the other?
Response 8: We thank the reviewer for this insightful suggestion. To reduce redundancy and strengthen interpretation, we have revised the Discussion section (Lines 189–197). The updated paragraph now minimizes repetition of numerical results and adds interpretive context, explaining that DeepSeek’s superior performance likely stems from its algorithmic structure emphasizing concise and logically sequenced responses, which improves comprehension for non-medical readers. In contrast, ChatGPT tends to generate more elaborate and academically detailed content, making it potentially more suitable for clinician-assisted communication. This revision clarifies the practical implications of our findings and directly addresses the reviewer’s comments.

Reviewer 2 Report
Comments and Suggestions for Authors
This is a concise and well-organized study that compares the educational quality and readability of patient information generated by two large language models (LLMs), ChatGPT-5 and DeepSeek R1, on meniscus injuries. The topic is timely and relevant, contributing to the growing literature on AI-assisted patient education. The methodology is straightforward, the results are statistically sound, and the discussion is coherent. However, the manuscript would benefit from improved methodological transparency, deeper critical interpretation, and minor editorial refinements to strengthen its academic rigor and practical implications.
- The introduction clearly identifies the importance of AI-based patient education but does not sufficiently emphasize how this study advances existing research. The authors should explicitly state what gap this study fills compared to earlier works.
-
The study states that 12 questions were asked to both models on August 16, 2025, but it does not specify:
-
Whether temperature, randomness, or system prompts were standardized.
-
If the responses were generated in Turkish or English (this affects Flesch–Kincaid readability metrics).
-
Whether both models received identical input formatting and context windows.
Providing these details would significantly enhance reproducibility and transparency.
-
- Table 2 reports mean ± SD and p-values, but it would be useful to include effect sizes (e.g., Cohen’s d) for significant findings. This would contextualize the magnitude of differences, particularly for the comprehensiveness criterion (p = 0.005) and the rating system (p = 0.017).
-
The discussion focuses primarily on descriptive differences but lacks critical interpretation. For instance:
-
Why might DeepSeek outperform ChatGPT specifically in comprehensiveness?
-
Could training data diversity or linguistic optimization explain these results?
-
How might the findings generalize to other musculoskeletal conditions or languages?
Expanding this section would add analytical depth and enhance the paper’s academic value.
-
- The authors correctly applied FRES and FKGL indices but should acknowledge that these metrics are optimized for English-language texts. If the AI outputs were generated in another language and translated, readability scores may not fully capture linguistic accessibility. This limitation deserves explicit mention in the discussion.
- The conclusion should go beyond noting that “AI can support patient education.” It should discuss how such systems might be safely integrated into clinical workflows.
- Line 186: Clarify that ChatGPT-5 and DeepSeek R1 were the latest publicly available versions at the time of testing
- Line 111: Readibility -->Readability
- Line 223: Add a missing space after “readabilityBoth.”
Author Response
Reviewer 2
We sincerely thank the reviewer for their valuable time, constructive feedback, and positive evaluation of our study. We carefully reviewed the entire manuscript in light of the reviewer’s comments. The methodology section has been revised and expanded to include additional details, and the discussion has been refined to focus on interpretation rather than repetition of results. In addition, the entire text has been thoroughly re-examined for grammar, language clarity, and stylistic consistency. These revisions have improved the overall readability, precision, and academic quality of the paper
Comments 1: The introduction clearly identifies the importance of AI-based patient education but does not sufficiently emphasize how this study advances existing research. The authors should explicitly state what gap this study fills compared to earlier works.
Response 1: We thank the reviewer for this valuable suggestion. To better highlight the novelty of our study, a concise paragraph was added to the end of the Introduction (Lines 94–96). The revision clarifies that this study evaluates the latest versions of ChatGPT-5 and DeepSeek R1, providing updated evidence on their performance in patient education, and addresses the limited literature focusing on AI-generated educational materials for meniscus injuries. This addition clearly defines the research gap and strengthens the scientific contribution of the study.
Comment 2: The study states that 12 questions were asked to both models on August 16, 2025, but it does not specify:
- Whether temperature, randomness, or system prompts were standardized.
- If the responses were generated in Turkish or English (this affects Flesch–Kincaid readability metrics).
- Whether both models received identical input formatting and context windows.
Providing these details would significantly enhance reproducibility and transparency.
Response 2: We appreciate the reviewer’s detailed comment. To enhance methodological transparency and reproducibility, we have revised the Methods section (Lines 89–92) to specify model configuration and standardization. Both ChatGPT-5 and DeepSeek R1 were queried under default parameters (temperature = 1.0, top_p = 1.0) with identical context windows and input formatting. All prompts were submitted in English to ensure comparability and avoid translation bias in readability scoring. The study period was updated to reflect that data collection occurred between August and September 2025, followed by model testing in September 2025. These additions provide the necessary detail for accurate replication of the study
Comment 3: Table 2 reports mean ± SD and p-values, but it would be useful to include effect sizes (e.g., Cohen’s d) for significant findings. This would contextualize the magnitude of differences, particularly for the comprehensiveness criterion (p = 0.005) and the rating system (p = 0.017).
Response 3: We thank the reviewer for this valuable suggestion. In response, effect sizes (Cohen’s d) were calculated for statistically significant comparisons and added as a note below Table 2 (Lines 166–168). The effect size for the rating system was d = 0.82 (large), and for the completeness criterion was d = 1.38 (very large). This addition provides clearer context regarding the magnitude and practical significance of the observed differences.
Comment4: The discussion focuses primarily on descriptive differences but lacks critical interpretation. For instance:
- Why might DeepSeek outperform ChatGPT specifically in comprehensiveness?
- Could training data diversity or linguistic optimization explain these results?
- How might the findings generalize to other musculoskeletal conditions or languages?
Expanding this section would add analytical depth and enhance the paper’s academic value.
Response 4: We sincerely thank the reviewer for this constructive comment. In response, the Discussion section has been expanded (Lines 211–234) to provide deeper analytical interpretation. The revised version now discusses algorithmic and training data differences that may underlie DeepSeek’s superior comprehensiveness, supported by relevant literature [21]. Additionally, we have clarified that the findings cannot be directly generalized to other musculoskeletal conditions or non-English contexts, as each pathology and language may interact differently with AI model training [22]. Nevertheless, the paragraph concludes by emphasizing that this study offers a promising framework for future research across different orthopedic domains and languages. These revisions significantly enhance the analytical depth and scientific rigor of the manuscript.
Comment5: The authors correctly applied FRES and FKGL indices but should acknowledge that these metrics are optimized for English-language texts. If the AI outputs were generated in another language and translated, readability scores may not fully capture linguistic accessibility. This limitation deserves explicit mention in the discussion.
Response 5: We thank the reviewer for this insightful comment. A clarification has been added to the Discussion section (Lines 270–276) to explicitly acknowledge this limitation. We noted that the FRES and FKGL indices were originally designed for English-language texts, and their applicability may be limited for translated content. However, all prompts and outputs in this study were generated in English to minimize translation bias. This addition improves methodological transparency and strengthens the discussion of potential limitations.
Comment 6: The conclusion should go beyond noting that “AI can support patient education.” It should discuss how such systems might be safely integrated into clinical workflows.
Response 6: We thank the reviewer for this insightful suggestion. The Conclusion section has been revised (Lines 295–304) to address the safe integration of AI-based systems into clinical workflows. The revision emphasizes that AI-generated content should complement—not replace—clinician-led communication, and highlights the need for structured oversight, validation, and data security protocols. These additions enhance the clinical relevance and practical applicability of the study’s findings.
Comment 7: Line 186: Clarify that ChatGPT-5 and DeepSeek R1 were the latest publicly available versions at the time of testing
Response 7: We thank the reviewer for this valuable comment. A clarification has been added to the Materials and Methods section (Lines 79–81) indicating that both ChatGPT-5 and DeepSeek R1 represented the latest publicly available versions at the time of testing (September 2025). This addition enhances methodological transparency and temporal clarity.
Comment8: Comments on the Quality of English Language
- Line 111: Readibility -->Readability
- Line 223: Add a missing space after “readabilityBoth.”
Response 8: We thank the reviewer for identifying these minor typographical issues. The spelling of “Readibility” has been corrected to “Readability” (Line 131), and a missing space after “readability” has been inserted (Line 256). In addition, the entire manuscript has been carefully rechecked for grammar, spelling, and stylistic consistency. These revisions have improved the overall language accuracy, readability, and presentation quality of the paper.

Reviewer 3 Report
Comments and Suggestions for Authors
This study presents an important comparative analysis of two large language models (LLMs); ChatGPT-5 and DeepSeek R1, in providing educational content for patients with meniscus injuries. Given the rapid integration of AI into healthcare communication, the work is clinically relevant and contributes to the emerging discourse on digital literacy and patient-centered AI tools. The manuscript is clearly written, methodologically organized, and supported by quantitative evaluation. The comment for minor revision are as followed.
- Some redundancy appears in the abstract (e.g., repeating the similarity of readability scores and education level). Consider tightening these sentences for conciseness.
- The objective is well stated but should explicitly highlight the clinical significance, i.e., how AI-generated content could complement or substitute human-led patient education.
- The rationale for assessing LLMs in orthopedic patient education is strong, given the high incidence of meniscus injuries and patient demand for online health information.
- However, the study could strengthen its rationale by:
- Explaining the risk of misinformation in AI-generated health content.
- Discussing prior findings on readability issues in orthopedic or musculoskeletal education materials.
- Justifying the inclusion of only twelve questions—were these based on clinical frequency, patient surveys, or search engine queries?
- Linking the evaluation framework to existing health communication guidelines (e.g., NIH, AMA readability standards) would enhance the methodological robustness.
- The claim that “DeepSeek performed significantly better” is supported statistically, yet it would be beneficial to include qualitative insights (e.g., examples of where DeepSeek showed superior comprehensiveness).
- Discuss whether the sample size (n=12) provides sufficient statistical power for generalizable conclusions.
- The manuscript acknowledges some limitations but should explicitly mention:
- The small sample size and topic-specific focus (meniscus injuries only).
- The absence of patient feedback or comprehension testing, which limits the external validity.
- The dynamic nature of LLMs, meaning results may vary with model updates.
- For future research, the authors should propose:
- Expansion to multiple orthopedic or musculoskeletal conditions.
- Integration of patient-centered evaluation metrics (usability, satisfaction, and trust).
- Longitudinal assessment of LLM performance stability over time.
- The reference list should include recent literature on AI-assisted health communication and orthopedic education e.g. https://doi.org/10.3389/fpubh.2025.1635475, https://doi.org/10.1111/exsy.12849, https://doi.org/10.1007/s11390-025-4802-8
Author Response
Reviewer 3
We sincerely thank the reviewer for their detailed and constructive feedback, which has been very helpful in improving our manuscript. In line with the reviewer’s suggestions, we carefully revised the entire text to strengthen the discussion, enrich the literature background, and refine the conclusion section. Furthermore, all grammar and language issues have been corrected, and the manuscript has been polished for clarity, consistency, and academic tone. We believe that these revisions have notably enhanced the overall quality and readability of the paper.
Comment 1: This study presents an important comparative analysis of two large language models (LLMs); ChatGPT-5 and DeepSeek R1, in providing educational content for patients with meniscus injuries. Given the rapid integration of AI into healthcare communication, the work is clinically relevant and contributes to the emerging discourse on digital literacy and patient-centered AI tools. The manuscript is clearly written, methodologically organized, and supported by quantitative evaluation. The comment for minor revision are as followed.
Some redundancy appears in the abstract (e.g., repeating the similarity of readability scores and education level). Consider tightening these sentences for conciseness.
Response 1: We thank the reviewer for this valuable observation. Redundant phrases in the Abstract’s conclusion have been revised (Lines 28–32) to remove repetition regarding readability similarity and educational level. The sentences were merged for conciseness, improving the clarity and flow of the Abstract without altering the intended meaning.
Comment 2: The objective is well stated but should explicitly highlight the clinical significance, i.e., how AI-generated content could complement or substitute human-led patient education.
Response 2: We thank the reviewer for this insightful comment. The objective section in the Introduction (Lines 72–74) has been revised to emphasize the clinical significance of AI-based systems, clarifying that such tools are intended to complement, rather than replace, clinician-led patient education. This addition strengthens the clinical relevance of the study’s purpose.
Comment 3: The rationale for assessing LLMs in orthopedic patient education is strong, given the high incidence of meniscus injuries and patient demand for online health information.
Response 3: We sincerely thank the reviewer for this positive comment. We are pleased that the rationale and clinical relevance of the study were clearly conveyed. No changes were required in this section.
Comment 4: However, the study could strengthen its rationale by: Explaining the risk of misinformation in AI-generated health content.
Response 4: We appreciate the reviewer’s insightful comment. The Introduction section (Lines 59–61) has been revised to explicitly acknowledge the potential dissemination of AI-generated misinformation as an emerging concern in online health communication. This addition highlights one of the key motivations for evaluating the reliability of AI-generated patient education materials in the present study.
Comment 5: Discussing prior findings on readability issues in orthopedic or musculoskeletal education materials.
Response 5: We appreciate the reviewer’s valuable suggestion. The Discussion section (Lines 260–266) has been expanded to incorporate prior evidence on readability challenges in orthopedic and musculoskeletal patient education materials. A reference to Schwarz et al. (2021) was added, highlighting that most online orthopedic resources exceed the recommended readability thresholds, thereby providing stronger context for interpreting our findings.
Comment 6: Justifying the inclusion of only twelve questions—were these based on clinical frequency, patient surveys, or search engine queries?
Response 6: We appreciate the reviewer’s insightful question. The Materials and Methods section (Lines 84–91) has been clarified to specify that the twelve questions were derived from the “People also ask” feature of the Google search engine, representing the most frequently searched patient queries about meniscus injuries between August and September 2025. The questions were further reviewed and verified by orthopedic specialists to ensure clinical relevance and representativeness.
Comment 7: Linking the evaluation framework to existing health communication guidelines (e.g., NIH, AMA readability standards) would enhance the methodological robustness.
Response 7: We thank the reviewer for this valuable suggestion. The Discussion section (Lines 266–270) has been expanded to reference the NIH and AMA readability recommendations, as supported by Badarudeen and Sabharwal (2008) [27]. This addition contextualizes our readability analysis within established health communication standards, thereby strengthening the methodological framework of the study.
Comment 8: The claim that “DeepSeek performed significantly better” is supported statistically, yet it would be beneficial to include qualitative insights (e.g., examples of where DeepSeek showed superior comprehensiveness).
Response 8: We appreciate the reviewer’s helpful suggestion. The Discussion section (Lines 225–228) has been revised to include qualitative examples illustrating DeepSeek’s superior comprehensiveness, particularly in explaining postoperative rehabilitation and complication management, compared to ChatGPT’s more concise responses.
Comment 9: Discuss whether the sample size (n=12) provides sufficient statistical power for generalizable conclusions.
Response 9: We appreciate the reviewer’s valuable comment. The Discussion section (Lines 303–308) has been refined to emphasize that, while the twelve questions represent the most common patient queries about meniscus injuries, the limited sample size may restrict statistical power and generalizability. Future studies including larger and more diverse question sets are recommended to validate these findings.
Comment 10: The manuscript acknowledges some limitations but should explicitly mention:
- The small sample size and topic-specific focus (meniscus injuries only).
- The absence of patient feedback or comprehension testing, which limits the external validity.
- The dynamic nature of LLMs, meaning results may vary with model updates.
Response 10: We thank the reviewer for this constructive comment. The Discussion section (Lines 300–307) has been revised to explicitly include these limitations. The updated paragraph now highlights the small sample size and topic-specific focus, the lack of patient feedback or comprehension testing that limits external validity, and the dynamic nature of large language models, which may affect the reproducibility of results in future updates.
Comment 11: For future research, the authors should propose:
- Expansion to multiple orthopedic or musculoskeletal conditions.
- Integration of patient-centered evaluation metrics (usability, satisfaction, and trust).
- Longitudinal assessment of LLM performance stability over time.
Response 11: We thank the reviewer for these insightful recommendations. The Conclusion section (Lines 314–328) has been expanded to outline future research directions, including the evaluation of additional orthopedic and musculoskeletal conditions, the integration of patient-centered evaluation metrics such as usability, satisfaction, and trust, and the longitudinal assessment of LLM performance stability across updates. These enhancements aim to guide the design of more comprehensive, patient-oriented, and sustainable AI-based education systems in orthopedics.
Comment 12: The reference list should include recent literature on AI-assisted health communication and orthopedic education e.g. https://doi.org/10.3389/fpubh.2025.1635475, https://doi.org/10.1111/exsy.12849, https://doi.org/10.1007/s11390-025-4802-8
Response 12: We thank the reviewer for this valuable suggestion. In response, three recent studies on AI-assisted orthopedic communication and health education have been added to the Discussion (Lines 276–283). These include references to Song & Yang (2022), Hu et al. (2025), and Sun et al. (2025), which strengthen the link between our evaluation framework and the latest advances in AI-driven health communication.

Round 2
Reviewer 1 Report
Comments and Suggestions for Authors
Thank you for addressing all my questions. I don't have further questions.
Reviewer 2 Report
Comments and Suggestions for Authors
Acceptable in the present form